# Storing Carbon in Forest Biomass and Wood Products in Poland—Energy and Climate Perspective

Zbigniew W. Kundzewicz [ID], Janusz Olejnik [ID], Marek Urbaniak [ID] and Klaudia Ziemblińska *[ID]

Meteorology Lab., Department of Construction and Geoengineering, Faculty of Environmental Engineering and Mechanical Engineering, Poznan University of Life Sciences, 60637 Poznan, Poland; zbigniew.kundzewicz@up.poznan.pl (Z.W.K.); janusz.olejnik@up.poznan.pl (J.O.); marek.urbaniak@up.poznan.pl (M.U.)
* Correspondence: klaudiaziem@wp.pl; Tel.: +48-61-8466533

**Abstract:** Huge amounts of carbon being sequestered in forest ecosystems make them an important land carbon sink at the global scale. Their ability to withdraw carbon dioxide ($CO_2$) from the atmosphere, whose concentration is gradually increasing due to anthropogenic emissions, renders them important natural climate-mitigation solutions. The urgent need for transition from high to zero net emission on country, continental, and global scales, to slow down the warming to an acceptable level, calls for the analysis of different economic sectors' roles in reaching that ambitious goal. Here, we examine changes in $CO_2$ emission and sequestration rates during recent decades focusing on the coal-dominated energy sector and Land Use, Land-Use Change, and Forestry (LULUCF) as well as wood production at the country level. The main purpose of the presented study is to examine the potential of storing carbon in standing forest biomass and wood products in Poland as well as the impact of disturbances. The ratio of LULUCF absorption of $CO_2$ to its emission in Poland has ranged from about 1% in 1992 to over 15% in 2005. From a climate-change mitigation point of view, the main challenge is how to maximize the rate and the duration of $CO_2$ withdrawal from the atmosphere by its storage in forest biomass and wood products. Enhancing carbon sequestration and storage in forest biomass, via sustainable and smart forestry, is considered to be a nature-based climate solution. However, not only forests but also wood-processing industries should be included as important contributors to climate-change mitigation, since harvested wood products substitute materials like concrete, metal, and plastic, which have a higher carbon footprint. The energy perspective of the paper embraces two aspects. First, $CO_2$ sequestration in forests and subsequently in harvested wood products, is an effective strategy to offset a part of national $CO_2$ emissions, resulting largely from fossil fuel burning for energy-production purposes. Second, wood as biomass is a renewable energy source itself, which played an important role in sustaining energy security for many individual citizens of Poland during the unusual conditions of winter 2022/2023, with a scarce coal supply.

**Keywords:** climate change; wood products; forest biomass; fuel wood

## 1. Introduction

Is it commonly known that coal plays a dominant role in national energy production in Poland. For many decades, the country has been a major coal producer as well. In 1979, coal production was nearly as high as 180 million metric tons a year. However, over recent decades, domestic production of coal has been gradually decreasing to ca. 77 million tons, slightly over 55 million tons, and about 53 million tons, in 2010, 2021, and 2022, respectively. In 2021, Poland was ranked the ninth-largest coal and lignite producer in the world and the second-largest in the European Union (EU) [1]. In the past, hard coal was produced in Poland mainly for domestic purposes but also for export to gain convertible currencies that were scarce and badly needed for the Polish economy before the great change in the political and economic system in 1989—switching from the communism system and

centrally planned economy to capitalism and market economy. However, Poland has gradually become a net importer of hard coal, with Russian Federation being the main source of import (responsible for 90% of the total import in 2020). In 2021, hard coal imports reached nearly 12.5 million tons. As elsewhere in the world, also in Poland coal production in 2021 expanded compared to the pandemic year 2020, indicating a rebound in demand. However, contrary to what was expected, hard coal production in Poland did not increase from 2021 to 2022, despite the turbulence due to the missing coal supply from Russia as a result of sanctions imposed upon the Russian Federation after it invaded Ukraine.

It is worth noting that besides energy and the economy, there are also meaningful linkages between coal and such systems and sectors as policy, employment, environment, climate, and health. Figure 1 illustrates the connections that the authors consider to be important for the scope of this paper, even if many more links can be envisaged. Some non-trivial links in Figure 1 merit interpretation. For instance, there are several paths from coal to health. Coal has been an essential energy source for heating purposes in the cold season. This is represented by the coal–energy–health path. The coal–environment–health path has to do with air pollution caused by burning fossils like hard coal and especially lignite, which affects morbidity and mortality. Several tens of thousands of deaths per year recorded in Poland are attributable to poor air quality. According to a report issued by the European Environment Agency, exposure to fine particulate matter caused 36,530 additional deaths in Poland in 2020. Thus, it was estimated that relative to the population number, Poland is among EU countries that are most affected by $PM_{2.5}$ air pollution, after Bulgaria, Romania, Croatia, and Hungary [2]. The coal–climate–health path embraces increased morbidity and mortality due to weather extremes enhanced by the warming (for which fossil fuel burning is responsible), therein heat-waves in particular. There is also an important coal–employment–health linkage. For many decades hard coal industry has been and still is a significant employer in Poland, with 77,449 employees (status of 31 January 2022) [3]. Miners suffer from profession-related morbidity and fatal mining disasters still occur. Except for the first link from coal via energy to health mentioned above (coal used for heating providing appropriate thermal conditions in winter), the other three paths end up with adverse health effects.

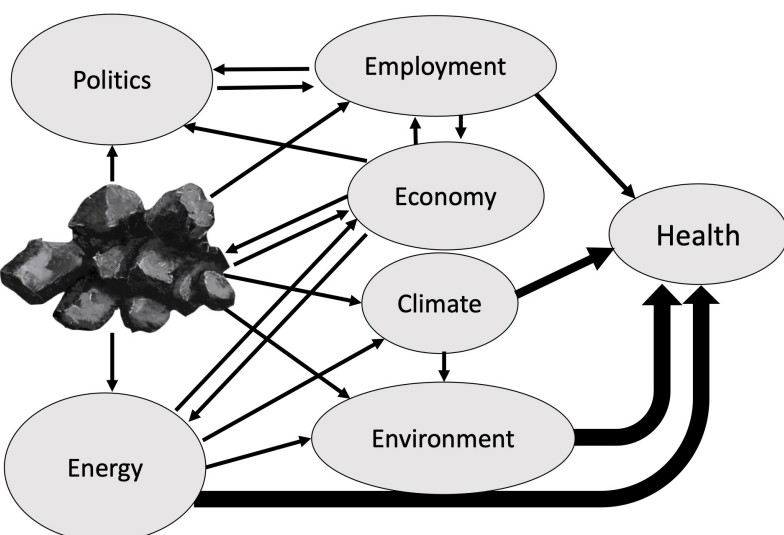

**Figure 1.** Coal is connected to multiple systems and sectors.

Due to their carbon sequestration (both in biomass and in the soil), terrestrial ecosystems constitute, globally, the second principal carbon sink after the ocean. It is estimated that in total they absorb about 30% of global anthropogenic $CO_2$ emissions [4,5]. To curb global warming, it is necessary to further enhance carbon sequestration but most importantly to reduce $CO_2$ emissions (by saving energy and setting on renewables) at the truly

global scale and as fast as possible. Therefore, every country, in particular countries like Poland, where the energy system largely depends on coal, has a substantial role to play.

In the presented paper, we examine forests' climate-change mitigation role by analyzing patterns of carbon storage in standing wood biomass and harvested wood products in Poland. In subsequent sections, we tackle two groups of issues: changes in atmospheric $CO_2$ emissions at the country level and the path to decarbonization in Poland on the background of global and EU climate policy. Next, the trends and disturbances in carbon sequestration by living forest biomass are analyzed and discussed. Finally, we recognized the main climate-change mitigation challenges from the forestry and wood production viewpoint: is it possible to increase the rate and duration of carbon storage in living wood biomass and harvested wood products?

## 2. Changes in Carbon Dioxide Emissions

The global human $CO_2$ emissions (with cement production included) have continued to rise over many decades (Figure 2). The trend is sustained while absolute values are even higher when LULUCF (Land Use, Land-Use Change, and Forestry) sector is included. The rate of increase has varied throughout decades, with the minimum growth rate being 0.9% per year in the 1990s and the maximum of 4.3% per year in the 1960s. Most importantly, in the last decade 2010–2020 global emissions more than trebled in comparison to the decadal value in the 1960s even though many "save-the-climate" initiatives have been launched [5]. The episodes of a temporary decrease in global emissions were linked to two oil crises in 1973–1975 and 1979–1982 [6], and later to the collapse of the Soviet Union and the onset of independence of satellite Eastern European countries (including Poland) in 1989–1992, the global financial crisis in 2008–2009, and finally the recent COVID-19 pandemic, resulting in a substantial drop in global emissions from 2019 to 2020. In 2021, the global $CO_2$ emissions exceeded 40 Gt $CO_2$ $yr^{-1}$, with the highest annual national emissions recorded in China (nearly 12.5 Gt $CO_2$ $yr^{-1}$). The average annual $CO_2$ emission rate per capita reached the value of 4.81 t globally, 6.25 t in the EU, and nearly 8.5 t in Poland [7]. Figure 2 also illustrates the concurrent time series of global Gross Domestic Product (GDP) growth. It is clear that, in relative terms, the $CO_2$ emission growth was initially faster than the global GDP growth for several decades after 1960, while in more recent decades it is GDP that increase more substantially than emission rates.

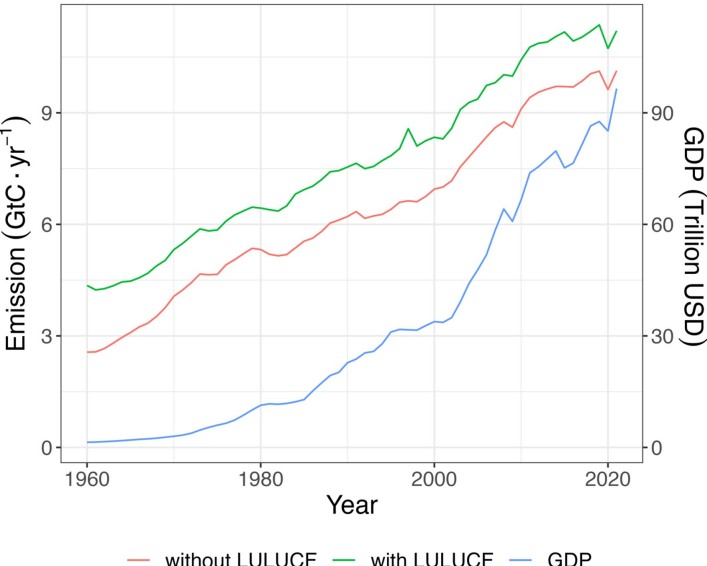

**Figure 2.** Time series of global $CO_2$ emissions, without LULUCF (red) and with LULUCF (green; data from Global Carbon Project [5]) and global GDP growth (blue; data from World Bank [8]).

In Poland, $CO_2$ is by far the most significant greenhouse gas (GHG) both regarding its emission and absorption. However, the trend of national $CO_2$ emissions in Poland in the last few decades (Figure 3) considerably differed from the average global curve presented above (Figure 2). This is mostly the effect of a massive decrease of $CO_2$ emissions following the systemic (political and economic) change commencing in 1989. The transition rendered the highly energy-consuming industries unprofitable and many of them went bankrupt. As a result, both national production and $CO_2$ emission decreased sharply, with the accompanying unemployment rate increase. However, in a couple of years, Poland commenced the recovery path, so that GDP gradually increased, while $CO_2$ emissions continued to decrease, due to the improvement of energy efficiency. Hence, Poland has proudly reported a decoupling between the economic growth and $CO_2$ emissions, unlike the global society. In 2020, national net $CO_2$ emissions in Poland were equal to 303.5 Mt (ca. 0.083 GtC), which is 35.7% lower than in the base year 1988 (472 Mt = 0.128 GtC), if LULUCF (Land Use, Land-Use Change, and Forestry) was not included in the calculation, cf. Figure 3. If LULUCF was included in the analysis, the national $CO_2$ emissions decreased from 450.7 Mt $CO_2$ eq. in 1988 to even 280.6 Mt $CO_2$ eq. in 2020, cf. Figure 3 [9]. The year 1988 was selected by Poland as the base year, in contrast to other countries, where the base year was typically assumed to be 1990. Since $CO_2$ emissions in 1988 reached their maximum values in Poland, exceeding those reported in 1990, thus the obtained emission reductions relative to the selected base year were more impressive. Total national greenhouse gas (GHG) emissions in Poland decreased from 579.4 Mt $CO_2$ eq. (LULUCF excluded) and 560.1 Mt $CO_2$ eq. (LULUCF included) in 1988 to, respectively, 376 Mt $CO_2$ eq. and 355.2 Mt $CO_2$ eq. in 2020 [9]. Fast economic growth in Poland from 2010 to 2019 was expressed by the GDP increase of as much as 38%. Furthermore, in the last pre-pandemic year (2019) Poland's economic growth rate was 4.7%, which is more than three times higher in comparison to the EU average (1.5%). Decoupling of energy production (and resulting $CO_2$ emission) from economic growth was possible mainly because of improvements in the energy efficiency of Poland's economy (total final energy consumption divided by domestic product) that decreased from 79 tons of oil equivalent (toe) per million USD in 2010 to 61 toe in 2019 [10]. However, the outbreak of the COVID-19 pandemic caused a decrease in Poland's economy as well as in energy production from 2019 to 2020, leading to a notable reduction of GHG emissions, largely driven by decreasing emissions related to transport (lockdown effect). However, total GHG emissions rebounded in 2021, because the energy demand returned and the share of coal-fired installations increased.

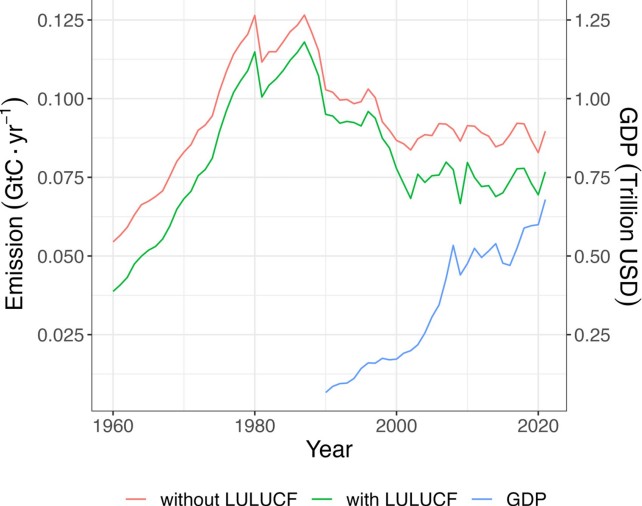

**Figure 3.** National $CO_2$ emissions in Poland during the period 1988–2020, with LULUCF (green) and without LULUCF (red) in GtC yr$^{-1}$ (recalculated from Mt $CO_2$ eq.—data source: Poland's National Inventory Report 2022 [9]) and national GDP growth (blue; data from World Bank [8]).

Figure 3 does not cover the pre-1989 GDP values, simply because of the lack of homogeneity of assessments, with clear discontinuity related to systemic change in the socio-economic system. In addition, before the political change, Polish currency was not convertible, so setting the proper exchange rate would be difficult. The visible drop in $CO_2$ emissions in 1980–1982, coinciding with the peaceful advance of the Solidarity Trade Union movement, can be interpreted as a result of a drop in production caused by massive protests against the party and the governmental policies.

The data compiled by IEA [10] for Poland show that, recently, partitioning of $CO_2$ emissions from particular fuels combustion in 2020, was: coal (hard coal and lignite) 57.7%, oil 28.3%, natural gas 12.0%, non-renewable waste 2.0%, and from respective sectors: electricity and heat generation 43.5%, transport 22.6%, industry 20.1%, buildings 13.8%. The coal-based energy sector is mostly responsible for GHG emissions in Poland.

## 3. Path to Decarbonization of Energy Sector in Poland vs. Global and EU Climate Policy

As previously stated, the global $CO_2$ emissions keep growing, despite long-term, advanced, global climate policies and international negotiations led at the Conferences of the Parties (COPs) of the United Nations Framework Convention on Climate Change (UNFCCC) that are held every year, gathering tens of thousands of participants. Unfortunately, their tangible results have been quite disappointing so far. The principal net result is opposite to what was intended—there is an overwhelming increase in global anthropogenic $CO_2$ emissions (Figure 2). In 2015, political leaders of most countries around the globe signed the so-called Paris Agreement introduced at the 21st UNFCCC Conference of the Parties (COP-21) to curb the intensification of the greenhouse effect and to reduce global warming [11]. The goal of the Paris Agreement ratified by nearly 200 nations, is to take steps towards efficient $CO_2$ emission reduction so that global warming is limited to well below 2 °C (preferably to 1.5 °C) above the pre-industrial level until the end of the 21st century. Unfortunately, climate-mitigation pledges made so far by national governments across the globe do not augur well for confining the warming to the range agreed in Paris. Moreover, implementation of these pledges has not been satisfactory so far [12]. At present, there are obvious risks of breaching not only the 1.5 °C guardrail but even the 2 °C line. After the last two COPs (COP 26 in 2021 in Glasgow, UK, and COP 27 in 2022, in Sharm el Sheik, Egypt) the conclusion emerged that while it must be attempted to mitigate global warming to at least below 2 °C (preferably to 1.5 °C), one should be prepared to adapt to a 3 °C warming [13]. Even though many countries (e.g., the EU and the USA) have reduced their GHG emissions, there has been a rapid increase in China, India, and many other countries. In brief, there is a global hiatus between the recognized science-based aspirations and the actual politics-driven actions. In order not to exceed 1.5 °C (2 °C) temperature rise, it is necessary to reduce net GHG emissions by half already in 2030s (2040s), relative to 2019, and then further to zero in 2050s (2070s). The value of zero net GHG emissions implies that $CO_2$ sequestration should compensate residual emissions in their entirety. Pathways that lead to the Paris Agreement targets require fast near-term transformation generating high upfront transition costs while auguring long-term economic gains including benefits in avoiding adverse climate-change impacts. An efficient climate policy depends on changes in the technological, economic, environmental, socio-cultural, and institutional dimensions that lead to emission reduction in several sectors and systems, but foremost in energy, buildings, industry, land use, and land cover.

In contrast to many countries in the world, the European Union (EU) follows the ambitious emission reduction path set in Paris Agreement in an exemplary way. In March 2020, the EU announced a long-term strategy aiming to achieve EU-wide carbon neutrality by 2050. In December 2020, the EU increased the 2030 GHG emissions reduction target from 40% to 55%, so that the new *"Fit for 55"* plan requires a 55% reduction of GHG emissions by 2030, relative to 1990. A legislation package backing the *"Fit for 55"* plan was adopted by the European Commission in July 2021, embracing regulations related to energy, land use, transport, and financial instruments [14]. Furthermore, the EU has been developing 2030

targets for renewable sources and energy efficiency, so that the new EU-wide emissions reduction target is supported.

There is no doubt that it will be difficult (if not impossible given the current trends and rates) for Poland to fulfill the international climate policy targets, even though the country has been largely expanding the usage of renewable energy sources. The share of coal in the energy mix declined in Poland already in the 2010s and has been gradually decreasing since then. This holds for total energy supply, electricity generation, and district heating [10].

Annual coal consumption in Poland has been decreasing (Figure 4), mainly due to decreasing demand in the power plants, being the main consumer of coal in the country. Consumption in other sectors was smaller and did not show any clear trends during this period. A drop in the pandemic year 2020 and an increase in 2021 is visible (Figure 4).

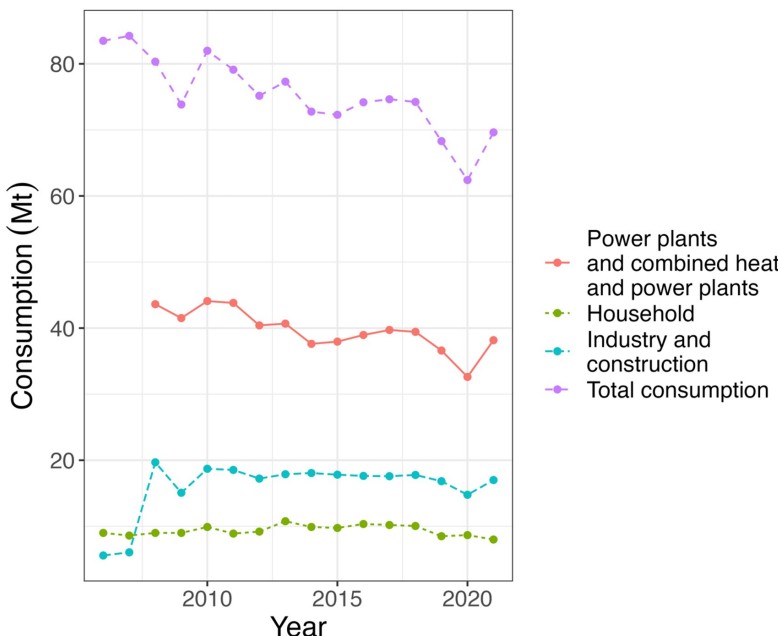

**Figure 4.** Coal consumption in selected sectors in Poland during the last two decades (data from Statistical Yearbook of the Republic of Poland 2022 [15]).

In 2021, Poland was still the second-largest contributor (after Germany) to the EU27's total fossil $CO_2$ emissions, with a contribution of 11.6% [7]. The existing balance of electricity [15] shows supply (equal to use) of 195 TWh in 2021 (as compared to 164 TWh in 2010), therein 180 TWh produced from domestic sources. The latter figure can be disaggregated into 141 TWh from public thermal power plants and the rest (ca. 40 TWh) from renewables, i.e., hydroelectric, wind, and solar photovoltaic (PV) power plants (as opposed to only 6.3 TWh in 2010). This illustrates that the supply of electricity from renewable energy sources has considerably increased, in both absolute and relative terms.

The energy mix for electricity generation in Poland in 2020 was still dominated by fossil fuels, even though the reduction of the carbon footprint in the 2010s was substantial. There has been notable success in efficient energy source transition. The total installed capacity of renewable energy systems constitutes 30.8% of all electricity sources in Poland [16], with wind over land being the dominating renewable source. It is worth mentioning that the very dynamic development of wind farms on land in Poland was virtually frozen in 2016 by a restrictive regulation passed in that year. Fortunately, only after seven years, in 2023, it was replaced by a less restrictive legal act, so that further development of wind farms on land is likely in Poland. It is also worth mentioning that Poland is one of the fastest growing photovoltaics (PV) markets in the EU and its PV capacity dramatically increased, from 0.2 GW in 2016 to 7.7 GW in 2021. Most of the increase refers to small-scale, distributed,

residential, PV systems. In 2018, Poland's hydrogen production exceeded 1 million tons (ranked third in the EU). There are also ambitious plans for energy decarbonization via offshore wind farms and the deployment of nuclear energy.

Two essential documents regarding energy and climate policy were issued in Poland in 2019–2021. First, the *National Energy and Climate Plan* [17] was adopted in 2019. Making such a plan was obligatory for all EU Member States. Next, in February 2021, the *Energy Policy for Poland until 2040* [18] came into force. Even though the necessity of gradual change in the energy mix towards the reduction of coal's share was recognized in these documents, coal was still projected to continue to play a key role in the energy sector for decades to come. National energy-mix goals, set in EPP2040, assume "to reduce the coal's share in electricity generation from the baseline of 70% in 2020 to below 56% by 2030 and 28% by 2040". However, the GHG emissions reduction specified in EPP2040 (30% relative to 1990) is well below the ambition of *"Fit for 55"* EU policy.

As depicted by links between coal and employment presented in Figure 1 and described in the Section 1, the coal industry continues to play a substantial part in the national labor market. The Polish government and the trade unions forged a social contract in May 2021, concluding the need for gradually phasing out domestic hard coal production and closing all of Poland's hard coal mines (except for coking-coal mines) by 2049. However, the contract does not include any targets to phase out lignite production or lignite-fired electricity production [10]. The decarbonization of the energy sector in Poland implies the reduction of employment in the coal mining industry. Thus, the contract guarantees social benefits. Workers in the hard coal sector will keep their job until retirement or receive a severance package. The contract also commits to supporting alternative economic development and economic transition in the main hard coal mining regions. However, it is recognized that the education level of miners is rather low while wage expectations are high. Indeed, miners have always been well paid in Poland. Therefore, it is quite difficult for them to find other jobs with comparably high salaries so the risk of unemployment is high [19]. For these reasons, the Polish government is forced to implement decarbonization policies that are not only environmentally friendly and cost-effective, but, most of all, socially acceptable. Another constraint on decarbonization is that it cannot jeopardize the national energy security that has largely relied on coal.

## 4. Carbon Dioxide Sequestration in Biomass on Land

It is estimated that forests cover approximately 31% of the Earth's surface [20] and contain 662 Gt of carbon, which is more than half of the total carbon stock in soils and vegetation biomass in general.

Grassi et al. [21,22] estimated that globally LULUCF sector was a net sink of 1.6 Gt $CO_2$ $yr^{-1}$ in 2000–2020, while the forest sink alone was four times greater (6.4 Gt $CO_2$ $yr^{-1}$). However, Grassi et al. [21–23] also indicated large differences in estimates between country reports and scientific (model-based) studies as well as a large range of estimates of carbon fluxes from LULUCF. This is to say that the uncertainty of estimations is still considerable.

Forecasting the future potential, Anderegg et al. [4] assessed that by 2030 forest-based strategies might provide up to 7 Gt $CO_2$ eq. of climate-change mitigation per year, globally, at a carbon price of 100 USD per 1 ton of $CO_2$ eq. This is by far the largest potential of natural climate solutions.

Existing European Union (EU) legislation includes the LULUCF sector in the Paris Agreement goals [24], where forests play the main $CO_2$ sink role. Nabuurs et al. [25] assessed that the overall annual climate-change mitigation effect of forests in the EU via contributions of the forest carbon sink, material substitution, and energy substitution could amount to 0.569 Gt $CO_2$ $yr^{-1}$ that was approx. 13% of the total EU $CO_2$ emissions in 2015. This was promising so it can be expected that under the EU *Green Deal* policy and, in particular, Union's *Biodiversity and Forest Strategy*, there will be even more emphasis on forests.

Already in 2017, *climate-smart forestry* was proposed in Europe, tackling multiple policy goals. Nabuurs et al. [26] postulated that it would potentially achieve more than the European Commission's (EC) legislative proposal of July 2016, incorporating GHG emissions and removals due to LULUCF into the 2030 Climate and Energy Framework. It was estimated that with the right set of incentives at the Union's and individual Member States' levels, the EU has the potential to achieve an additional combined mitigation impact through climate-smart forestry of 0.441 Gt $CO_2$ $yr^{-1}$ by 2050.

However, early signs of saturation of carbon sink in European forests were noted since the early 2010s [27]. After centuries of stock decline and deforestation, commencing in and lasting throughout the Middle Ages, vegetation rebound was prevalent since the 1950s. Forests in Europe have recovered in area and growing stock, forming a persistent carbon sink that was projected to continue. Meanwhile, three indicators of European forest biomass signaled emerging problems of carbon sink saturation: the stem volume increment rate has been decreasing, the land-use change has been intensifying, and the natural disturbance frequency and intensity has been increasing.

In Poland, forests cover 92.6 thousand $km^2$, which is 29.6% of the total land area [28]. They are thus the main $CO_2$ sink in the country. Since World War II, the forest area in Poland has been gradually increasing and an increase in harvested wood production has been recorded simultaneously. The average growing stock of Polish forests is now 254 $m^3$ $ha^{-1}$ [9]. Since most forests grow on poor sandy soils, coniferous species prevail, constituting 68.6% of tree species composition. Climatic and habitat conditions in Poland are favorable for pine species, with dominant Scots pine—*Pinus sylvestris* that occupies 58.6% of the forest area in the country.

There is an obligation under Articles 3.3 and 3.4 of the Kyoto Protocol of the UNFCCC to report the national balance of emissions and removals of greenhouse gases, therein activities related to LULUCF. Activities related to afforestation/reforestation and forest management are considered to result in net $CO_2$ absorption (sometimes called *negative emissions*). However, the removals of atmospheric $CO_2$ from the LULUCF sector in Poland largely differed in individual years, ranging from as low as 2–10 Mt in 1992–1994 to as high as above 50 Mt in 2004–2005 (Figure 5). The $CO_2$ absorption decline was caused by natural disturbances that had a considerable impact on the level of carbon accumulation in forests, such as wildfires (1992), severe droughts (1992–1993, 2015, and 2018–2019), as well as the most severe wind damages in 100-year Polish State Forests' history that were recorded in August of 2017. As a consequence, considerable changes in the photosynthesis rate and dead wood decomposition occurred. The quantitative assessment of the impact of such events, as well as following reforestation procedures, on the carbon balance of Scots pine forest in Poland was presented in Ziemblinska et al. [29]. Among the recognized reasons explaining a decrease in $CO_2$ removals are also the effects of tree stand aging affecting tree size and shape, biomass allocation, as well as resulting changes in allometric relationships [9].

Data presented in Figure 5 seems to corroborate the statement of Anderegg et al. [4] that climate-driven risks to forest stability from fire, drought, biotic agents, floods, wind-breaks, and other disturbances may fundamentally compromise forest carbon sinks. Due to the occurrence of severe drought in 1992 and 2018 the LULUCF absorption rate, together with its $CO_2$ emission offset value (LULUCF $CO_2$ absorption/$CO_2$ emission total ratio) significantly dropped. Furthermore, an illustration of inter-annual variability of net ecosystem production (NEP = $CO_2$ absorption − $CO_2$ emission), estimated based on real-time observations using the eddy-covariance method at a single measuring tower in Tuczno forest (West Pomeranian Voivodship, Poland) [30,31], given in Figure 6 seems to support these findings too. The data used to produce Figure 6 comes from the only long-term direct measurements of GHG exchange between Scots pine (*Pinus sylvestris*) forest and the atmosphere with high temporal resolution (measurement at the frequency of 10–20 Hz averaged to 30–min $CO_2$ fluxes) in Poland. Inter-annual variability of net carbon exchange can be very high and reflects, among other factors, the cumulative impact of meteorological

conditions, especially the effect of water-related disturbances. As derived from Figure 6, the difference between annual NEP values at the end of the year (annual NEP total which, if positive as in the presented example, can be treated as a total annual $CO_2$ sequestration rate) for the "best" year on record, 2009, equal to 20.76 t $CO_2$ ha$^{-1}$ and the "worst" (dry) year, 2019, amounting to 7.62 t $CO_2$ ha$^{-1}$, is indeed striking.

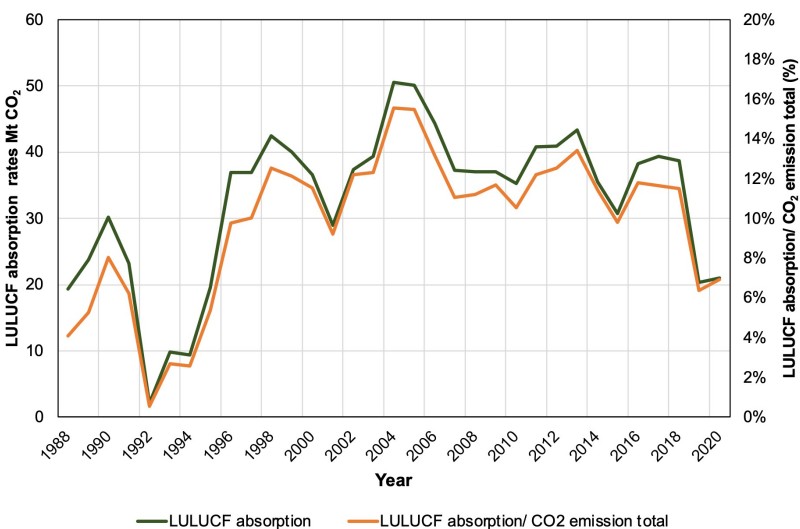

**Figure 5.** Time series of total annual LULUCF absorption of $CO_2$ and a ratio of LULUCF absorption of $CO_2$ to total $CO_2$ emission in Poland during the period 1988–2020 in Mt $CO_2$ (data from Poland's National Inventory Report 2022 [9]).

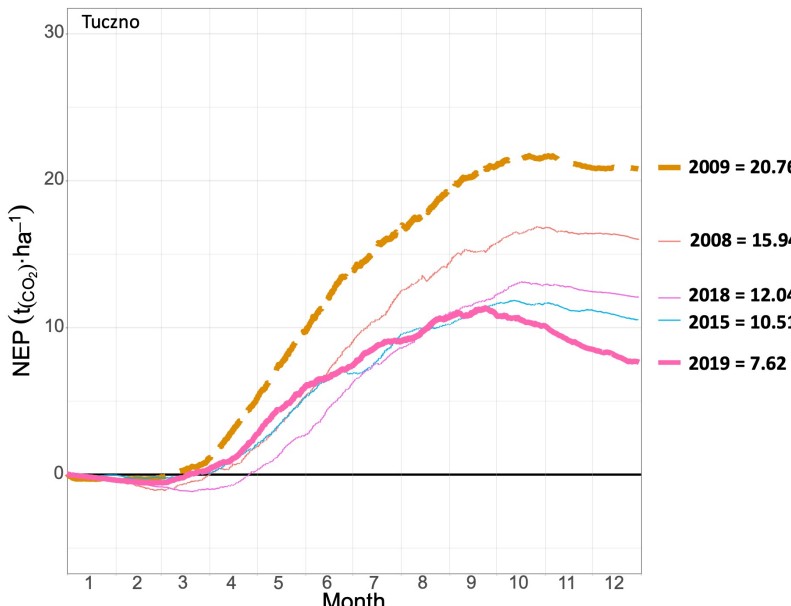

**Figure 6.** Course of cumulative net ecosystem production fluxes (NEP = balance between absorbed and emitted $CO_2$) in the Tuczno forest site (West Pomeranian Voivodship, Poland) for selected years [30]. Data were collected in the framework of the project funded by State Forests—National Forest Holding in Poland (see Funding).

Boysen et al. [32] noted that to fulfill the UNFCCC Paris Agreement, parties should not only massively reduce near-term GHG emissions but also extract the already emitted $CO_2$ from the atmosphere as much as possible. Specifically, this could refer to harnessing

the sequestration abilities of vegetation and the establishment of fast-growing trees and grass plantations in combination with carbon utilization. Simultaneously, the same authors stated that "biomass plantations with subsequent carbon immobilization are likely unable to "repair" insufficient emission reduction policies without compromising food production and biosphere functioning" [32]. Thus, even though such deployment can be regarded as promising, there are considerable uncertainties about its carbon sequestration potential and possible side effects. Much space and water would be needed. Nevertheless, even if this strategy of sequestering more carbon is not a viable alternative to much-needed massive emission reduction, it holds promise for supporting climate-change mitigation actions.

Building on the same topic, Ho [33] explicitly warned of false hopes, stating that "decarbonization must come first, or carbon removal will be next to useless". He analyzed carbon dioxide removal (CDR) needed for offsetting the residual emissions that are responsible for the *net* term in *net-zero emission*. Residual emissions are projected to remain high in sectors that are 'hard to abate', such as aviation and shipping. In such sectors there is no way to achieve a zero-emission goal, hence the CDR is really necessary for achieving the net-zero-emission target, globally. Buck et al. [34] found that residual GHG emissions at net-zero emissions could be of the order of 18% of current emissions for UNFCCC Annex I countries. The CDR approaches can be divided into planting or maintaining vegetation to enhance natural carbon sequestration or removing $CO_2$ from the atmosphere by other means. The first approach is well recognized, yet it is largely insufficient in light of huge and gradually increasing emissions. The second approach, if at all feasible, is only in its infancy phase. There are small-scale demonstrations of Direct Air Capture (DAC) technologies, sucking $CO_2$ out of the atmosphere by chemical reactions. Creating four DAC hubs in the USA is planned but these solutions, even if powered by renewable sources of energy, would not save the global climate now, when GHG emissions are high, and further increasing. If successful scaling up is possible (that is itself problematic—what works well in a laboratory may not necessarily work at a large field scale), indeed they might be of importance if we successfully do the bulk of decarbonization first. There are challenges and caveats—the need to minimize land use, energy consumption, and environmental risk. An apt summary of the issues discussed above is given by Ho [33]: "Humanity has never removed an atmospheric pollutant at a … [large–term added] scale—we have only ever shut down the source and let nature do the clearing up".

## 5. Maximization of Volume and Duration of Carbon Storage in Forest Biomass and Wood Products

Maximization of volume and duration of sequestered carbon storage in standing wood biomass and harvested wood products is a challenge [35]. We strongly believe that wood-processing industries should also be regarded as important contributors to climate-change mitigation. The timespan of carbon storage in harvested wood products ranges from months (or less) in the case of biomass used for pulp production, to many decades for construction wood in buildings [36].

There is a dual mechanism for the contribution of wood products to climate-change mitigation. First, they store, for possibly long periods of time, carbon withdrawn from the atmosphere by trees. Second, they substitute other materials which would have been used instead, like concrete, steel, and plastic, which have a higher carbon footprint, since their production requires higher energy input and results in higher total $CO_2$ emission.

The policy of promoting the use of wood supplies from local sources and species would also serve biodiversity by helping protect precious tropical forests from massive deforestation (but may simultaneously weaken the economic growth of some countries of the Global South). Furthermore, there is a high carbon footprint associated with the long-distance transport of tropical wood from rainforests to Europe, North America, Japan, etc.

One can also maximize the carbon storage in wood products by maximizing carbon fluxes entering the pool of wood products (inflow) and minimizing those leaving that pool (outflow). The pool of carbon stored in wood products can be increased by enhancing the

allocation of harvested wood to long-lived wood products or by increasing the lifetime of these products and by increasing their recycling rate. Enhanced and improved use of wood products is thus crucial [37].

There are several categories of raw wood materials, such as firewood, paper wood, industrial round wood, sawn wood, wood-based panels, paper and paperboard, wood pulp, and recovered paper. The use of construction wood is of particular interest. This can refer to entire wooden houses, or building carpentry: roof truss, floors, stairs, doors, and windows, as well as furniture, and garden wood.

In the first centuries of the history of Poland, whose origin dates to the decade of the 990s, timber was the main material used for building houses. However, wooden cities burned easily, e.g., during turbulent times of wars frequently taking place in the Polish lands. Therefore, wooden constructions were considered to be much more fragile and inferior to those made of stone and brick. Therefore, the substantial legacy of the Polish king, Kazimierz Wielki (Casimir the Great), who lived from 1310 to 1370 and reigned from 1333 to 1370, was that, as worded by the 15th-century Polish historian, Jan Długosz, "he found Poland made of wood and left it made of brick". Among large-scale climate-change mitigation efforts based on harvested wood products, buildings made of wood offer considerable advantages of long-term carbon storage as well as using wood as a substitute for non-wood structural materials with a higher carbon footprint. Further intensive growth of the global population and urbanization rate is very likely and would be accompanied by ever-increasing demand for new housing and commercial buildings. Production of such constructive materials as masonry, cement, concrete, steel, and glass generates GHG emissions that are much greater than those necessary for the construction of wooden buildings. Thus, Churkina et al. [38] advocated massive usage of engineered timber in constructing buildings. Harvested wood products used in the construction of large buildings are typically laminated from smaller boards or lamella to form large structural components in CLT (cross-laminated timber) panels and glulam (glue-laminated) beams. As noted by Kozłowski [39], thanks to appropriate treatment (removal of most of the hemicellulose and lignin from the wood by chemical methods, and then pressing the wood at high temperature and pressure), it is possible to transform wood into a material that is more durable and much lighter than steel.

It was advocated that "long-term lock-in of carbon on land" should be ensured and this can be achieved through increasing the durability of construction [38]. Among pre-conditions for achieving higher wood harvest levels while maintaining carbon storage in forests are advancing reforestation, preserving forest sustainability (via legal and political commitment), extending reserves in vulnerable forests of high biological value, adequate forest certification schemes, etc.

The wooden buildings are tall, presently. The record tallest mass timber building in the world in 2021 was the 25-story 86.56 m high Ascent apartment tower in Milwaukee, but many higher wooden buildings are either planned or under construction so this record is unlikely to stay for long. Indeed, as observed by Tollefson [40] "timber buildings are getting safer, stronger, and taller—and they could help to cool the planet". In Poland, in late 2022, a well-auguring MOD21 startup initiative was launched by the ERBUD group [41] aimed at the production of wooden modular constructions. Wooden modules, produced indoors, in a factory, can be assembled in various combinations to serve housing, public services (e.g., education and health establishments), and commerce (shops, offices, hotels, parlors, etc.). Among the advantages offered by wooden modular architecture are economy, speed, durability, sustainability, flexibility, and mobility.

In some circumstances, the wood consumption for heat generation can be also meaningful. Iordan et al. [42] examined wood energy consumption in three Nordic countries (Sweden, Finland, and Norway), where in 1960–2015 about 6.6 billion $m^3$ of wood were harvested from which 638 million $m^3$ (less than 10%) were used for energy production, while 2.3 billion tons of solid products were manufactured simultaneously. In Sweden, where district heating is dominating, bioenergy contributed by 23% to the total primary

energy supply, with about 85% coming from forest harvesting (logging and forest industrial residues). In Finland, where bioenergy generation is subsidized, wood-based fuels cover 88% of the total renewable energy generation. However, the above examples from Scandinavia, where the number of inhabitants is low and the forested area is large, cannot be upscaled to other European countries. In brief, burning wood cannot be generally seen as a viable substitute for burning fossil carbon.

For the last two decades total timber harvest reported by the State Forests – National Forest Holding (SFNFH), managing more than 80% of all forests in Poland, was ca. 40–50% less than estimated wood volume increments [43], i.e., a sufficient amount of trees' biomass was left to grow as the forest area in Poland was also gradually increasing. The amount of harvested wood (the majority of which was then available on the market) ranged between ca. 40 million m$^3$ in 2017 and 2018, to slightly more than 30 million m$^3$ around 2005. In general, until the pandemic year 2020, there was an increasing trend in total harvested timber (THT), which was accompanied by a decreasing total annual wood volume increment (AWI). Therefore, the AWI/THT ratio has recently increased to 63–67%. Meanwhile, an increasing tendency in foreign trade of wood and wood products has been observed [15], with the maximum value of import to and export from Poland in 2021 reaching 3.294 and 7.381 billion USD, respectively.

The course of total annual coniferous timber sold in Poland during the last 20 years showed a clear increasing trend, which resulted from an increase in both saw timber and pulpwood similarly, while sales of deciduous timber did not fluctuate much. A visible, sharp peak in all coniferous timber categories was recorded in 2018, which can be attributed to the huge wood supply following unprecedented damage in Polish forests, caused by a hurricane in the summer of 2017. There was an interesting pattern regarding fuelwood sales. Since 2005 it was steadily increasing, reaching the highest values in the post-windthrow 2018 year, equal to 1.43 and 1.80 million m$^3$ for coniferous and deciduous wood, respectively, and then dropping down to less than 1.15 and 1.46 million m$^3$ in the pandemic 2020 year. After the restrictions regarding coal import after the Russian invasion of Ukraine on 24 February 2022, the demand for wood in Poland as a substitute source of energy in individual households was so significant that the amount of sold fuelwood in that year was from 1.5 to almost twice higher than the records of previous years (Figure 7).

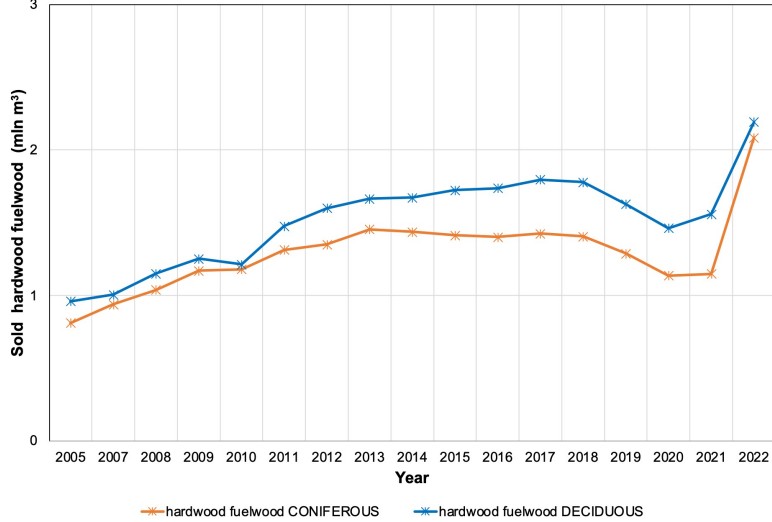

**Figure 7.** Hardwood fuelwood (coniferous and deciduous species) sale by the State Forests—National Forest Holding (SF-NFH) in Poland in 2005–2022 (data from Annual Economic and Financial Reports of State Forests National Forest Holding [43]).

In general, the energy perspective of the forestry and wood products sector in Poland embraces at least two aspects. First, $CO_2$ sequestration in forests and subsequently storing it in wood products is an effective strategy to offset a part of $CO_2$ emissions, largely caused by fossil fuel burning. Second, wood as biomass is a renewable energy source itself that played an important role from an individual citizen viewpoint in sustaining the energy security in Poland during the emergency of the cold season of 2022/2023 (Figure 7). The import of Russian coal was halted after Russia's invasion of Ukraine and the prompt increase of domestic coal production in Poland throughout 2022 turned out to be impossible. As a result, there was a scarce coal supply in Poland—a country sitting on coal, where the energy system is dominated by coal. The two emergency options to improve short-term energy security in Poland in 2022 were as follows. First, coal was imported from overseas, including very remote places like Australia and South America. This was not a climate-friendly solution, because of the great carbon footprint generated by the long-distance transport of coal. Second, fuelwood use as derived from the sales rates increased considerably in 2022 (Figure 7). More firewood and even branches collected in forests were burned in individual households. Again, this emergency solution cannot be regarded as effective and especially because it is not climate-friendly. The newly collected wood was not seasoned; hence its energetic value was lower as much of the energy had to be wasted on evaporating water from the wood while burning.

However, replacing coal with wood as the fuel for heating (and—to less extent—cooking) alone in households in Poland is impossible, not to mention the absurd idea of replacing coal with wood in power plants. As shown in Figure 4, households in Poland use about 10 Mt of coal per year, corresponding to about 255 (range: 220–275) PJ of energy. This estimate was determined under the assumption of the average calorific values of coal to be 25.5 MJ/kg (range: 22–27.5 MJ/kg) [15]. To obtain the same amount of energy, it would be necessary to burn about 25 million m$^3$ of wood (range: 20–36 depending on the tree species and wood moisture). This number was determined with the assumption of a calorific value of 10,000 MJ/m$^3$ (range: 7000–13,000). Meanwhile, on average, less than 1.3 million m$^3$ of coniferous and 1.5 million m$^3$ of deciduous hardwood are used for heating purposes (Figure 7). This is an order of magnitude smaller than the estimated need. Hence, it is clear that such substitution is unrealistic because the current supply of firewood itself would be insufficient for substituting coal in households if the supply for other wood products is to be left unchanged. Furthermore, problems of air quality and smog occurrence would become even more severe, resulting from burning not only coal but also wood in domestic conditions.

For completeness, it is worthwhile to indicate more categories of wood products in Poland. For instance, Poland is the largest supplier of wood for furniture to the IKEA company [44], contributing 28%, which is much more than Lithuania (10%) and Sweden (9%). Poland has also been a major supplier of wood products for small garden architecture to European networks of department stores and supermarkets selling garden supplies. Adequate treatment of these products, granting satisfactory wood protection, provides three-fold benefits [45]:

1.  Economic (longer utilization of a wooden product, hence there is no need to buy a new one)
2.  Social (possibility of purchasing a durable product of good quality that may serve satisfactorily over a longer time), and
3.  Environmental (storing absorbed $CO_2$ in harvested wood products for a longer time).

It is a common practice to sell inexpensive wooden elements of small garden architecture, such as wooden palisades and garden sticks, as well as roll borders, within the fourth class of wood use (outdoors, in contact with the ground and/or freshwater). However, the low price is coupled with the low quality of wood treatment. Untreated or poorly treated pine wood products are likely to serve no longer than 2–3 years, depending on the conditions, such as product size, soil, humidity, shade, and vegetation impact. Hence, the benefits regarding the three areas mentioned above are in jeopardy. Microorganisms

decomposing wood cause prompt release of $CO_2$ to the atmosphere. The product life is short so it is necessary to buy a new product after a couple of years and install it again. Consequently, garden owners may be disappointed by a wooden product and may wish to replace it with a more durable one e.g., made of plastic, metal, or concrete, with a much higher carbon footprint.

A peculiar option for global climate-change mitigation, recognized in the literature [46], is a carbon sequestration strategy, in which harvested dead or living tree wood is buried in trenches or stowed away in above-ground shelters. Due to mostly anaerobic conditions, the decomposition rate of the buried wood would be low in comparison to open-air exposition. By cutting off the return pathway of $CO_2$ to the atmosphere, an effective carbon sink can be created. According to Zeng [46], this low-cost and low-tech technique could be a promising option for large-scale implementation. However, this approach has not been recognized in global strategies. Also, the direct conversion of roundwood into biochar does not seem to attract large attention. Perhaps the massive use of timber in building construction is a more convincing and economically viable way to store carbon.

## 6. Discussion and Conclusions

In this paper, we examined the energy and climate perspective of storing carbon in forest biomass and wood products in Poland, with global and European Union scales in the background. The essence of the climate perspective is that forest ecosystems provide natural climate-change mitigation by counteracting the ongoing global increase of $CO_2$ emissions from burning fossil fuels that lead to intensification of the greenhouse effect and warming. Forests withdraw $CO_2$ from the atmosphere, whose concentration has been gradually increasing, globally, and build carbon in its biomass, acting as a very important land carbon sink. Emission reduction has been globally agreed upon as an essential target for humankind, to avoid the adverse effects of dangerous levels of uncontrolled climate change and to curb the warming to an acceptable level.

The observed changes in $CO_2$ emission and sequestration rates in Poland, whose energy sector has been coal-dominated, were also presented. We analyzed the potential of storing carbon in standing forest biomass and wood products in the country, including the impact of disturbances. From a climate-change mitigation point of view, the main challenge is—how to maximize the rate and the duration of $CO_2$ withdrawal from the atmosphere by its storage in forest biomass and harvested wood products. However, early signs of saturation of carbon sink in European forests have been noted, related to the following emerging problems: decreasing stem volume increment, intensifying land-use change, and increasing frequency and intensity of natural disturbances. It was estimated that major climatic extremes account for 78% of changes in global gross primary productivity over the last 30 years, and severe droughts accounted for 60 to 90% of these phenomena. Fires account for about 12% of disturbances in forest ecosystems annually, but their share and importance in different parts of the world is different [4]. Existing projections from the Earth system model for the sustainability of $CO_2$ sequestration by the world's forests are probably too optimistic, increasing the need to reduce greenhouse gas emissions [47]. Other authors emphasize that if trees grow faster as a result of environmental changes that stimulate growth, they will either reach dimensions that qualify them for felling sooner, or they will go through their natural life faster according to the principle "grow fast-die young" [48]. In the end, the current chance of harvesting more wood based on the current forest biomass increase due to climate warming might be very short-term. As shown in Figure 5, the inter-annual variability of the absorption of $CO_2$ in the LULUCF sector to total $CO_2$ emission ratio in Poland is high and has ranged from the low value of about 1% in 1992 (year with massive drought and extensive wildfires) to over 15% in 2005. It is worth noting that annual wood increment (AWI) data has shown a clear downward trend (relative year-to-year AWI differences were negative) from 2018 onwards due to disturbances related to climate change (mainly drought), while total timber harvest only slightly increased. Therefore, most probably even if we want to acquire more because demand for harvested

wood and wood products is/will be growing (as proved by the example of the coal supply crisis), it might not be possible. So that the perspective "let's replace coal with native wood material", while maintaining its use in other sectors (construction wood, paper, sale to IKEA) at the same level, may also be impossible due to the deteriorating condition of forests caused by climate change.

Even if forests alone cannot prevent further global warming, they have played and are projected to play an even more important role in climate-change mitigation. It is becoming clear that after the bulk of effective decarbonization is accomplished, globally, via reduction of $CO_2$ emissions, forests will be indispensable for implementation of the adjective *net* in the *net-zero emissions* target.

Synergies between such objectives as climate-change mitigation and sustainable development should be sought, generating environmental, economic, and social co-benefits. Sustainable forest management assumes at least keeping, preferably increasing, the carbon stock in the global biomass. Practices of wood acquisition cannot conflict with sustainable forest management. A holistic look is necessary: a comprehensive balance of carbon, energy, and cost in the production processes, as well as thorough consideration of side effects whose costs and benefits should be internalized.

Strengthening ecosystem carbon sink is both a challenge and an opportunity for the forestry sector. Therein, the selection of proper rotation period and forest harvest intensity which leads to sustainability and does not result in biodiversity deterioration are among the major challenges.

Among large-scale climate-change mitigation efforts based on harvested wood products, buildings made of wood offer considerable advantages of long-term storage of carbon as well as substituting non-wood structural materials (masonry, cement, concrete, steel, and glass) with a considerably higher carbon footprint. Hence, the massive use of timber in building construction seems to be a convincing and economically viable way to store carbon in harvested wood products.

The tangible benefits of the present paper embrace two aspects. First, $CO_2$ sequestration in forests and subsequently in harvested wood products, is an effective strategy to offset a part of national $CO_2$ emissions in Poland, resulting largely from fossil fuel burning for energy-production purposes. Here the energy and climate perspectives have large common parts. Second, wood as biomass is a renewable energy source, so fuelwood may be a substitute for fossil fuels as a source for energy production. Fuelwood itself played an important role as an emergency solution during the unusual conditions of the cold season of 2022/2023, with scarce coal supply due to the elimination of imports from Russia. Availability of wood enhanced energy security from the viewpoint of many individual citizens of Poland in an emergency.

However, it should be strongly emphasized that, based on rough estimations, substituting coal with wood in Poland even in households alone (not to mention power plants and industry) is unrealistic, because the available supply of wood itself would be far too low. Additionally, the severity of problems related to air quality and smog occurrence would increase in the country.

**Author Contributions:** Conceptualization, Z.W.K.; methodology and data curation, M.U.; writing—original draft preparation, Z.W.K.; writing—review and editing, M.U., K.Z. and J.O.; visualization, M.U. and K.Z.; supervision, J.O.; technical manuscript preparation: K.Z. All authors have read and agreed to the published version of the manuscript.

**Funding:** The conducted research was supported by funding from the General Directorate of the State Forests, Warsaw, Poland (project name: LAS IV 31/2021/B; contract no. EZ.271.3.3.2021).

**Data Availability Statement:** All data shown in the paper in a graphical form were derived from publicly available sources (reports, published articles, book chapters, etc.), and correctly cited whenever used.

**Conflicts of Interest:** The authors declare no conflict of interest.

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
