# Peer review of "Storing Carbon in Forest Biomass and Wood Products in Poland—Energy and Climate Perspective"

_energies, doi:10.3390/en16155788_

Round 1
Reviewer 1 Report
Dear Authors,
The article submitted for review, "Carbon storage in forest biomass and wood products in Poland - an energy and climate perspective," aims to investigate the role of forests in mitigating climate change by analyzing patterns of carbon storage in standing biomass and harvested wood products in Poland.
After reading the article, I have the following comments and suggestions for improving it:
Structure of the article
I suggest improving the structure of the article according to the ENERGIES journal guidelines
The article is missing the Discussion and Materials and Method chapters.
Abstarkt
I propose to make corrections according to the journal's guidelines. What is the purpose of the research? What research methods were used? Please present the main conclusions.
Introduction
The introduction should be deepened with world literature. I suggest adding information about the current status of this research. Who has been involved so far. Why te badania są important?
Results
The results are presented and described in a very good way and are very interesting. They contribute to the value of the paper.
In Discussion chapter, the authors should discuss and explain the conclusions and results of the work more. What tangible benefits have been gained from conducting this study?
This would contribute to a high improvement of the work. The authors should compare their project and results with the results of similar studies on this topic from other countries in the world. .
Kind regards,
Reviewer
Author Response
Legend:
R – Reviewer’s comments
A – Authors’ reply
R: The article submitted for review, "Carbon storage in forest biomass and wood products in Poland - an energy and climate perspective," aims to investigate the role of forests in mitigating climate change by analyzing patterns of carbon storage in standing biomass and harvested wood products in Poland.
After reading the article, I have the following comments and suggestions for improving it:
Structure of the article.
I suggest improving the structure of the article according to the ENERGIES journal guidelines.
The article is missing the Discussion and Materials and Method chapters.
A: Our methodological approach is essentially based on construction of important, relevant, and novel graphs based on available data that one can find (even if not easily) in the public domain and our own NEP data, to support our expert narratives. To large extent, we wrote this paper from scratch, because we are not aware of any existing source item in the literature that covers the entirety of our material. Therefore, in order to avoid the obligation to have separate components labelled “Discussion” and “Materials and Methods” that should be components of a submission in the “Article” category, the authors selected another category - an “Essay”. It thus allows us to freely select its components. Since the paper has multiple foci, presentation of “Materials and Methods” in one place would not be optimal. However, elements of classic “Materials and Methods” section are distributed throughout the paper. To partly comply with Reviewer’s comments, we combined Discussion and Conclusions in the last section of the revised manuscript.
R: Abstract. I propose to make corrections according to the journal's guidelines. What is the purpose of the research? What research methods were used? Please present the main conclusions.
A: Thank you for these good ideas. We tried to address the purpose and the conclusions. Although, we do not think that mentioning methods in the abstract is necessary.
R: Introduction. The introduction should be deepened with world literature. I suggest adding information about the current status of this research. Who has been involved so far. Why te badania są important?
R: Discussion. The authors should compare their project and results with the results of similar studies on this topic from other countries in the world.
A: To the authors’ knowledge, there exist no paper in the literature covering the entirety of the topics spanned by our paper, at any scale and in any country. Certainly, there are many references addressing some elements of our paper. In response to Reviewers’ #1 and #3 comments, we thus tried to enrich our list of references from the original number of 44 to currently 48 references in the revised manuscript.
R: Results. The results are presented and described in a very good way and are very interesting. They contribute to the value of the paper.
A: The authors are flattered by these positive remarks.
R: In Discussion chapter, the authors should discuss and explain the conclusions and results of the work more. What tangible benefits have been gained from conducting this study?
This would contribute to a high improvement of the work.
A: The authors are really grateful for this constructive and useful advice. We tried to react to it in the last, concluding section, entitled Discussion and Conclusions now.
Reviewer 2 Report
Dear authors,
thank you for presenting this study/essay. The topic is interesting, but I have the following remarks:
- the beginning with information from the history is not well-structured.The aim of the paper is "perspective", so aiming for future. Some information about history that gives insights into Polish perspective are okay, but there is no need to present everything (does the reader really need to know all information about the "coal issue"? - I don´t think so);
- there are redundant information on other countries perspectives. I understand that authors tried to provide a comparison, but since the paper is an "essay" and about situation in Poland, some of the information are not important for the paper (e.g., the content of the Paris Agreement, generally known). Author also mix the information by presenting the situation in Poland and in EU (and/or in other countries) and make it hard for the reader to follow;
- there is often a mixture of presenting information on the international, European and national level. This is again not well structured;
- point 5 - why there is so much space devoted to the general information;
- I would also encourage authors to use more references from Poland.
Some of the sentences are hard to follow (too long or complicated sentences). I would suggest proof-reading by a native speaker.
Author Response
Legend:
R – Reviewer’s comments
A – Authors’ reply
R: Thank you for presenting this study/essay. The topic is interesting, but I have the following remarks:
- the beginning with information from the history is not well-structured. The aim of the paper is "perspective", so aiming for future. Some information about history that gives insights into Polish perspective are okay, but there is no need to present everything (does the reader really need to know all information about the "coal issue"? - I don´t think so);
A: The term perspective could mean at least three different things. It can refer to temporal perspective (as assumed by Reviewer #2), but it can also refer to spatial perspective (we tackle global, EU and national, i.e. Polish scales). As far as the temporal perspective is concerned, we stick to observations and evaluations related to past-to-present, rather than dealing with projections for the future. The title of our paper contains the term “energy and climate perspective” that primarily refers to the energy and climate viewpoint. We could change the title of the paper, replacing “energy and climate perspective” by “energy and climate viewpoint”, if needed.
R: - there are redundant information on other countries perspectives. I understand that authors tried to provide a comparison, but since the paper is an "essay" and about situation in Poland, some of the information are not important for the paper (e.g., the content of the Paris Agreement, generally known). Author also mix the information by presenting the situation in Poland and in EU (and/or in other countries) and make it hard for the reader to follow;
- there is often a mixture of presenting information on the international, European and national level. This is again not well structured;
A: The authors refer to Poland but on the background of the Globe and/or the EU. Actually, Reviewer #1 wished to see more information from beyond Poland. We shortened the part that refers to the Paris Agreement, as suggested by the Reviewer. However, we would like to emphasize that it was decided to describe it in details since we were not sure if all the potential readers of ENERGIES generally know the contents of the Paris Agreement.
R: - point 5 - why there is so much space devoted to the general information;
- I would also encourage authors to use more references from Poland.
A: We gave some general information for the convenience of the reader. Virtually, to our best knowledge, no one has approached the topic from the viewpoint taken in our paper, so that we believe that a general introduction might be of a great use. In section 5 we refer to six references from Poland and four references from other countries since in our opinion, there is a general scarcity of references from Poland. Furthermore, Reviewer #1 wished to see more information from other countries in the world.
R: Comments on the Quality of English Language: Some of the sentences are hard to follow (too long or complicated sentences). I would suggest proof-reading by a native speaker.
A: We have carefully read the paper and simplified several long and complex sentences. We also used a service of a native speaker who has proofread the paper.
Reviewer 3 Report
Dear authors,
The essay you have submitted provides interesting material. Due to the fact that it is an essay, I have not reviewed it as a scientific publication. I believe that with some minor methodological work, the presented publication could have been more like a scientific review paper.
The present analyses of historical facts and statistical data complement each other. They indicate a strong relationship in the creation of the energy policy of the Polish state. In my opinion, the essay is suitable for publication - please comment only on the information about timber exports from Poland - which I think is an important and objective fact to present in view of the growing demand. Especially in the paragraph about timber construction.
I am sending more detailed comments in the attached .PDF file.

Author Response
Legend:
R – Reviewer’s comments
A – Authors’ reply
R: The essay you have submitted provides interesting material. Due to the fact that it is an essay, I have not reviewed it as a scientific publication. I believe that with some minor methodological work, the presented publication could have been more like a scientific review paper.
A: We are quite satisfied from categorization of our paper as an essay. We tried to keep the material original and interesting to read, covering much of the ground (perhaps more than an article could do, because it would need a narrower focus). Actually, we do not feel the need to convert our essay into an article or a review paper, because then it would be obligatory to have separate sections on Material and Methods as well as a Discussion (we have now Discussion and Conclusions as the last section, though).
R: The present analyses of historical facts and statistical data complement each other. They indicate a strong relationship in the creation of the energy policy of the Polish state. In my opinion, the essay is suitable for publication - please comment only on the information about timber exports from Poland - which I think is an important and objective fact to present in view of the growing demand. Especially in the paragraph about timber construction.
A: We are very pleased to read this opinion expressed by the Reviewer. Indeed, we agree with this opinion. Thus, we added some current information on timber import/export from Poland as suggested: “Meanwhile, an increasing tendency in foreign trade of wood and wood products has been observed [15] with the maximum value of import to and export from Poland in 2021 reaching 3.294 and 7.381 billion USD, respectively.”- lines 460-462 in the revised manuscript
R: I am sending more detailed comments in the attached .PDF file
A: We have incorporated these detailed comments into the present version of the manuscript and respond to them in the sequel.
R: Line 25 However, there is no separate section on literature review - it was suggested adding more references to peer-reviewed publications here.
A: We do not have a separate section on literature review that is actually not obligatory even in an article and really optional in an essay. To our knowledge, there exist no paper in the world literature covering the entirety of the topics spanned by our paper, at any scale. Certainly, there exist many references addressing elements of our paper. However, in response to Reviewers #1 and #3, we tried to enrich our list of references from the original number of 44 references to the number of 48 references in the presented, revised manuscript.
R: Line 26 The term is too colloquial. It also misrepresents some very interesting research material.
A: Actually, we thought that the somewhat colloquial term in the first sentence encourages the reader to spend time on our essay, being a lighter read than an article. However, we follow the advice of Reviewer #3 and take the term “sits on coal” out, so that perhaps “very interesting research material” is represented better.
R: Line 28 The data presented relate to the important background issues of the paper. However, I have a suggestion for the authors to consider - wouldn't it be better to present them in a graph? I think it would have a significant impact.
A: It would be very easy to fulfil this suggestion, perhaps even presenting a complete time series from 1970s-1980s to present. We also adhere to the wisdom that a picture tells more than a thousand words. However, in our view, such a graph would come very early in the paper and would be of marginal importance for our paper. Perhaps conveying the message in one sentence is better here. If Reviewer #3 insists, we will install a graph.
R: Line 36 Not all readers of an international journal may be aware of the facts that are being written about - the suggestion is to be more specific.
A: Good point. Thank you. We make it abundantly clear now.
R: Line 48 To confirm the facts, I would suggest adding a source to the publication that explored the issue.
A: We do not know about other work where a figure similar to our Figure 1 was used. It is our original contribution stemming from our common-sense considerations rather than from any existing work that is known to us.
R: Line 53 Source missing.
A: The source comes in the next sentence in the text.
R: Line 88 The abbreviation is used for the first time in the main section of the work - suggests expanding it.
A: We have complied with the Reviewer’s comment accordingly
R: Line 105 Source missing.
A: We did not copy this sentence from any existing source. It is our original observation, based on the material presented and cited in this paper.
R: 108 The description is ok, perhaps worth repeating in the introduction?
A: Thank you for the suggestion. We have added it in the introduction.
R: Line 152 The historical facts and events described did take place, but with such expressions, in my opinion, it should be referenced as in the humanities - citing the primary sources.
A: We have added more primary references accordingly.
Round 2
Reviewer 1 Report
The article has been revised according to the reviewer's instructions.